# The Content of Selected Heavy Metals and Polycyclic Aromatic Hydrocarbons (PAHs) in PM$_{10}$ in Urban-Industrial Area

**Natalia Zioła * and Krzysztof Słaby**

Institute of Environmental Engineering of the Polish Academy of Sciences, 34 M. Skłodowska-Curie Str., 41-819 Zabrze, Poland; krzysztof.slaby@ipis.zabrze.pl

* Correspondence: natalia.ziola@ipis.zabrze.pl

**Abstract:** This research concerns the measurement of daily PM$_{10}$ concentrations and the assessment of heavy metals and polycyclic aromatic hydrocarbons (PAHs). The measurements were carried out in the urban-industrial area in southern Poland in the period from February to December 2019 (covering heating and non-heating seasons). The metal content of As, Cd, Pb, Ni, Co, Cr, Cu, Zn, V, was estimated using mass spectrometry with inductively excited plasma (ICP-MS), and that of Au and Mg using atomic emission spectrometry with induced plasma (ICP-OES). Analysis of selected PAHs (Naph, Acy, Ace, Fl, Phen, An, Fluo, Pyr, BaA, Chry, BbF, BkF, BaP, IcdP, DahA, BghiP) was performed using a gas chromatograph coupled with mass spectrometry (GC-MS). The share of the analyzed metals in PM$_{10}$ concentrations was ~1.37% in the entire measurement period, ~1.09% in the heating season and ~1.55% in the non-heating one. High concentrations of aluminum and chromium, observed over the measurement period, indicate the presence of strong anthropogenic sources of both metals. In the case of PAHs in PM$_{10}$, the average total share of the analyzed was ~1.25%, while this share slightly increased in the heating season (average: ~2.19%) compared to non-heating (average: ~0.63%).

**Keywords:** PM$_{10}$; heavy metals; PAHs; urban-industrial area; air quality

## 1. Introduction

### 1.1. Particulate Matter

Particulate matter (PM) is a dispersed mixture of three-phase aerosol, which is defined by the European Environment Agency (EAA) as a mixture of liquid and solid particles suspended in the air [1]. The most important parameter of these particles is their diameter, which has a significant effect on their physicochemical properties as well as the speed of gravitational free fall. There are four limits to the diameter of the particles; less than 0.1 μm—PM$_{0,1}$ (called ultra-fine dust), up to 1 μm—PM$_1$, up to 2.5 μm—PM$_{2.5}$, up to 10 μm—PM$_{10}$ and Total Suspended Particles (TSP). There is also a fraction with a diameter between 2.5 μm and 10 μm called PM$_c$—particulate matter coarse [2]. Research carried out in urban-industrial areas, related according to the day and the measuring period, shows that 68–99% of TSP is the PM$_{10}$ fraction, which contains about 60% of PM$_1$ fraction by mass [2,3]. This points to the PM$_{10}$ fraction as the most comprehensive for air dust monitoring. This is in accordance with the applicable Regulation of the European Parliament and of the Council on PM$_{10}$, known as dust passing through the sorting hole, specified in the reference method for the flow and measurement of PM$_{10}$ (EN 12341:2014-07), at 50% power limit for aerodynamics diameter up to 10 μm [4]. Atmospheric aerosol solids include, among others: minerals, elemental carbon, sulfates, nitrates, ammonia, trace elements (heavy metals), and organic applications (PAHs) [5]. Particulate matter

origin can have both primary source (natural and anthropogenic) and secondary source, when they are created in the atmosphere, as a result of transformations of particle precursors. The anthropogenic origins of the particles, which can have various sources, are crucial for understanding and elimination of air pollution. In highly developed European countries, significant dust emissions come from communication transport, and matches or exceeds dust emissions by the municipal and household sector [6]. The main source of $PM_{10}$ dust in the air in Poland is combustion processes not coming from industry, but mainly emissions from household hearth, where solid fuel such as coal, wood and biomass are used (under local oxygen deficiency in the furnace), which are almost entirely half of overall emissions. Combustion processes in the industry and traffic have lower impact on the total emission of particulate matters [7]. Negative effects of particulate matter on human health, in particular on respiratory and circulatory systems, have been notice many times. About 40% of deaths (2 million people per year) among EU inhabitants are caused by cardiovascular disease, especially hypertension, whose main determinant of occurrence is the harmful effect of particulate matters. A correlation is observed between the mortality of smokers and an increase of air pollution. With an increase of air pollution by 10 $\mu g \cdot m^{-3}$, the risk of dying from hypertension for a smoker increases by 113%. PAHs and heavy metals are especially harmful components of dust [8].

*1.2. Polycyclic Aromatic Hydrocarbons*

Polycyclic aromatic hydrocarbons (PAHs) are condensed polycyclic hydrocarbons containing from two to several aromatic rings without substituents. PAHs with two or three rings (naphthalene, acenaphthene, anthracene, fluorene, phenanthrene), are present in the air mainly in the gas phase, whereas those with four rings (fluoranthene, pyrene, chrysene) are present in both the gas phase and the aerosol, and those having five or more rings (benzo[a]pyrene, benzo[g,h,j]pyrene) are mainly condensed on suspended particulate matters. These compounds never occur individually; the presence of one of the compounds from this group indicates the presence of the others [9]. Monitoring of PAH content in the air is very important because of the toxic, carcinogenic and mutagenic effects on living organisms [10]. PAHs emitted to the atmosphere are produced in the process of pyrolysis and incomplete combustion of solid fuels, waste, plant residues, etc. with oxygen deficiency [11]. Such processes take place in the installation of solid fuel heating furnaces, central heating furnaces, tiled kitchens and fireplaces. The National Center for Emissions Balancing and Management (KOBIZE), calculating the percentage part of PAHs emissions in various economic sectors through the presence of four PAHs (benzo[a]pyrene, benzo[b]fluoranthene, benzo[k]fluoranthene, indeo[1,2,3-cd]pyrene), revealed the total percentage shares of PAH emitting sectors, and showed that domestic heating processes were the most important. The participation of these four PAHs was at the level of 84% of the total emissions of PAHs in Poland in 2017 [12]. It was also found that 16% of BaP emissions and 10% of the sum of four PAHs emissions comes from industrial processes, particularly coking process and aluminum production. Reporting of emissions of $PM_{10}$, these four aromatic hydrocarbons as well as heavy metals, is required in the coke industry as part of the European Pollutant Release and Transfer Register (E-PRTR) protocol [13]. In 2017, emissions from Poland of the four abovementioned PAHs constituted approximately 10% of the total PAHs emissions from European Union countries, which is the third place behind Portugal and Germany. Over the many years, however, a downward trend in PAHs emissions has been observed in Poland and Europe. Because the main production of PAHs is related to municipal and household emissions (heating of buildings), seasonality of emissions is observed. As the atmosphere temperature cools, emissions increase, and decrease when it's getting warmer. A relationship was also observed between the quantity of PAHs in the air and meteorological conditions, especially in areas where vehicle traffic was the main source of these pollutants [12]. The highest content of PAHs was observed mainly in winter, at low temperatures with low wind speeds and cloudy conditions. The increase in concentration of PAHs in an urban area is also favored by the increase in relative humidity and strong wind blowing from the directions where the communication routes are located. A significant impact of wind direction on the increase of PAH air pollution in an

urban-industrial area was noticed in the analysis of the impact of large emission sources (such as industry or energy). Studies also show a relationship between an increase in PAH concentration and temperature inversion [14].

### 1.3. Heavy Metals

Heavy metals are a group of elements to which more and more attention is being paid by scientists. Heavy metals are characterized by density higher than 4.5 g·cm$^{-3}$, good thermal and electrical conductivity in liquid and solid states, they are not transparent and have gloss [15]. They also show toxic properties caused by their ability to accumulate in tissue and organs. These elements can get through the skin, can be inhaled and consumed with plant and animal products. Some of the heavy metals (As, Zn, Cd, Cu, Hg) can cause immediate acute poisoning, others (As, Zn, Cd, Cr, Cu, Hg, Pb, Sn, Co, Ni, Mn, Se, Fe and Ag) cause chronic conditions [16]. Non-biodegradability is the reason for their long-term persistence in the environment [17]. Due to their high durability and toxicity, heavy metals play an important role in the pollution of the environment. The percentage share of emissions from various sectors of the Polish economy is different for selected heavy metals, however, the manufacturing processes sector and the energy production and transformation sector prevail in the production of pollution [18]. Until recently, the most significant source of heavy metal emissions to the atmosphere in Poland was smelters (iron, steel and non-ferrous metallurgy) [17]. Heavy metals have a specific feature that distinguishes them from other pollutants produced by the industry. In the process of dedusting exhaust gases they are enriched with heavy metals, due to which their concentration in the dust emitted along with the exhaust gases from the chimney is relatively higher than in the dust contained in exhaust gases before dedusting them. In recent years, due to the modification of production processes, the use of wet and fabric dust collectors for dedusting of exhaust gases, and the hermitization of processes, emissions from metallurgy have significantly decreased.

Many studies have shown a different nature of the composition of atmospheric pollution in various urbanization structures, especially in urban and industrial areas. The aim of this study was a qualitative and quantitative analysis of PM$_{10}$ aerosol in terms of heavy metal content and PAHs on the example of the city of Skawina, and a comparison of the results obtained with other urban-industrial centers.

## 2. Methodology

### 2.1. Study Area

The measurements were carried out in the city of Skawina (49.9° N, 19.8° E), the seat of the urban-rural commune, located in the Krakow County, in the Lesser Poland Voivodship, in the south of Poland (Figure 1). The city itself, with an area of 20.50 km$^2$, has 23,182 inhabitants [19].

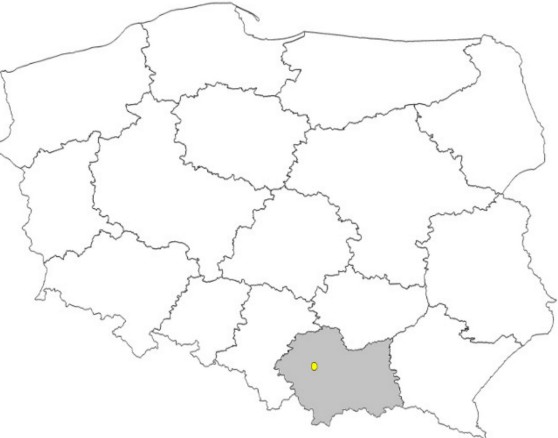

**Figure 1.** Map of Poland and the sampling location in Skawina (49.9° N, 19.8° E).

According to the physico-geographical division of Poland, the city is located in the Vistula River valley on the border of two mesoregions: Skawina Ditch and Wieliczka Foothills, on the border of two geological regions: the Carpathian Foredeep and the Outer Carpathians [20]. The basin nature of the area has a significant effect on climatic conditions, mainly unfavorable (frequent temperature inversions, stagnation of cooled air, high relative humidity, frost) [21]. The dominant direction of inflow of air masses is west and southwest. Unfavorable meteorological conditions, especially the lack of wind and inversion, significantly contribute to the accumulation of pollutants near the earth's surface and the formation of smog, as a consequence of exceeding the concentration of permissible concentrations of pollutants, especially $PM_{10}$. The monitoring station (belonging to the Institute of Environmental Engineering of the Polish Academy of Sciences in Zabrze) was located in the western part of the city, in the central part of the zone comprising several factories carrying out various industrial activities. The most important factors affecting the state of air pollution in Skawina are local and point emissions, mainly domestic emissions (in the heating season, suspended dust concentrations reach the highest values in the country), industrial emissions (95% dusts and 99% gases on the scale of the Krakow County), transport pollutions (national road No. 44 and voivodship road No. 953), inflow emissions (neighborhood of the city of Krakow) [21]. According to WHO reports, the city is at the forefront of the most polluted cities in the European Union [22]. On the basis of the European Union directives: (2008/50/EC and 2004/107/EC), and national regulations related to the directives, annual air quality assessment reports in individual voivodships are carried out by Voivodship Inspectorates for Environmental Protection [23]. These reports include 12 substances listed, including: $PM_{10}$, heavy metals (lead, arsenic, cadmium, nickel) contained in $PM_{10}$, and benzo(a)pyrene from the group of polycyclic aromatic hydrocarbons associated with $PM_{10}$. The Lesser Poland Voivodship has been characterized into three zones: the Krakow Agglomeration, the city of Tarnów, and the Lesser Poland Zone (the rest of the Lesser Poland Voivodeship) in which the city of Skawina is located. For each zone, the quality class of the allowable or target pollution standard is defined, which is the average of measurements at monitoring stations throughout the zone. Class A is not exceeding the allowable or target pollution standard levels, and class C is above these levels. The Lesser Poland Zone received class C for $PM_{10}$ and B[a]P in $PM_{10}$. For the indicated heavy metals it received class A. This assessment is inaccurate for the city of Skawina by averaging the measurements for the entire zone.

### 2.2. Equipment and Analytical Procedure

$PM_{10}$ analysis included a series of measurements carried out in the period from 23 February to 31 December 2019. This period included the heating seasons (23 February–31 March and 01 October–31 December), as well as the non-heating season (1 April–30 September ). These periods are contractual and refer to average atmospheric temperatures when district heating is required or redundant. This selection of periods gives an insight into the relationship between the seasons and the concentration of air pollutants. Determination results for metal content and PAHs came from 45 weekly samples, collected continuously in a daily cycle. The collection was carried out on quartz filters with a diameter of 47 mm (Whatman QMA) by gravimetric method, using the reference low-flow sampler µPNS LVS16 (MCZ) with a cooled receiving compartment. This sampler met the requirements of European regulations regarding the collection of $PM_{10}$, contained in the PN-EN 12341:2014-07 standard: "Air quality. Determination of particulate matter $PM_{10}$. Reference method and field test procedure to demonstrate the equivalence of the measurement method used with the reference method". According to this method, the sampler was calibrated for air flow of 2.30 $m^3 \cdot h^{-1}$. Before weighing, the clean filters were conditioned for 48 hours in laminar chamber, at stable temperature and humidity. Each filter was weighed on a Mettler Tolledo AT-20 microbalance (range 0.001 mg–0.1 mg). The weighted, clear filters were placed in holders and inserted into special transport and measuring cassettes in order to protect them during transport against any possible contamination in the air. After 14-day measurement cycles, the filters were transported in cassettes, and again conditioned in the laboratory for 48 h before being weighed on microbalance. The filters were cut in half before the actual analysis (half for metal analysis,

half for PAHs analysis). Determination of metal content (As, Cd, Pb, Ni, Co, Cr, Cu, Zn, V) was done using mass spectrometry with inductively excited plasma (ICP-MS), whereas Au and Mg using atomic emission spectrometry with induced plasma (ICP-OES). Compound samples were mineralized prior to analysis. Qualitative and quantitative analysis of selected PAHs was performed using a gas chromatograph coupled with mass spectrometry (GC-MS). Compound samples were subjected to extraction and concentration prior to analysis. The DAVIS Vintage Pro 2 automatic weather station was used to obtain primary meteorological data (wind speed and direction, temperature, humidity, total radiation, precipitation height).

## 3. Results

The presented results relate to the content of heavy metals and polycyclic aromatic hydrocarbons (PAHs) in $PM_{10}$.

The course of daily $PM_{10}$ concentrations was presented in statistical description (Table 1) and graphical form (Figures 2 and 3). Daily $PM_{10}$ concentrations were averaged for the entire measurement period, separately for the heating (23 February–31 March 2019 and 1 October–31 December 2019) and non-heating (1 April–30 September 2019) seasons, and individual months.

**Table 1.** Descriptive statistics of daily $PM_{10}$ concentrations ($\mu g \cdot m^{-3}$) from entire measurement period (23 February–31 December 2019), and by heating (23 February–31 March 2019 and 1 October–31 December 2019) and non-heating (1 April–30 September 2019) seasons.

| Specification | Ave ± SD | Min | Max | Median | Limit Value Exceeded [1] |
|---|---|---|---|---|---|
| Entire measuring period | 42.28 ± 29.66 | 4.03 (10/3/19) | 214.59 (30/11/19) | 32.25 | 85 |
| Heating season | 52.68 ± 34.70 | 4.03 (10/3/19) | 214.59 (30/11/19) | 43.54 | 51 |
| Non-heating season | 34.97 ± 22.96 | 7.49 (31/8/19) | 109.84 (3/4/19) | 25.42 | 34 |
| February 2019 | 68.35 ± 22.37 | 42.04 (23/2/19) | 108.68 (27/2/19) | 66.36 | 5 |
| March 2019 | 45.13 ± 26.75 | 4.03 (10/3/19) | 105.62 (22/3/19) | 40.04 | 10 |
| April 2019 | 68.23 ± 27.14 | 25.80 (21/4/19) | 109.84 (3/4/19) | 70.34 | 17 |
| Mai 2019 | 22.64 ± 7.68 | 8.95 (23/5/19) | 40.51 (8/5/19) | 21.61 | 0 |
| June 2019 | 43.57 ± 23.90 | 16.35 (20/6/19) | 99.95 (5/6/19) | 37.04 | 11 |
| July 2019 | 28.80 ± 10.79 | 19.40 (8/7/19) | 71.32 (1/7/19) | 24.71 | 1 |
| August 2019 | 27.30 ± 13.58 | 7.49 (31/8/19) | 54.85 (22/8/19) | 22.72 | 4 |
| September 2019 | 23.28 ± 12.02 | 11.85 (8/9/19) | 67.97 (24/9/19) | 20.79 | 1 |
| October 2019 | 40.75 ± 18.41 | 15.99 (1/10/19) | 98.39 (31/10/19) | 40.44 | 8 |
| November 2019 | 59.63 ± 42.26 | 15.57 (29/11/19) | 214.59 (30/11/19) | 46.35 | 13 |
| December 2019 | 60.80 ± 42.50 | 8.35 (28/12/19) | 173.05 (12/12/19) | 42.28 | 15 |

[1] The limit value for the average daily $PM_{10}$ concentration is 50 $\mu g \cdot m^{-3}$ and may not be exceeded more than 35 times a calendar year [4].

Over the entire measurement period (23 February–31 December 2019) 298 daily $PM_{10}$ concentrations were obtained. Therefore, this study met the time coverage requirements for continuous and periodic measurements [4].

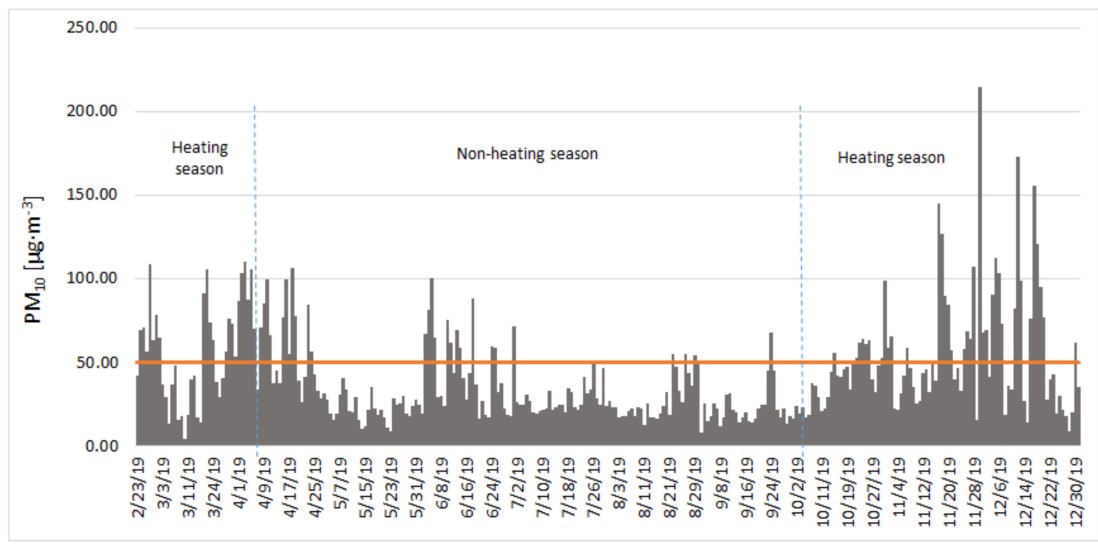

**Figure 2.** Series of daily concentrations of particulate matter $PM_{10}$—measurement period from 23 February to 31 December 2019 (orange line indicates the limit value for the average daily concentration of particulate matter $PM_{10}$ [4]).

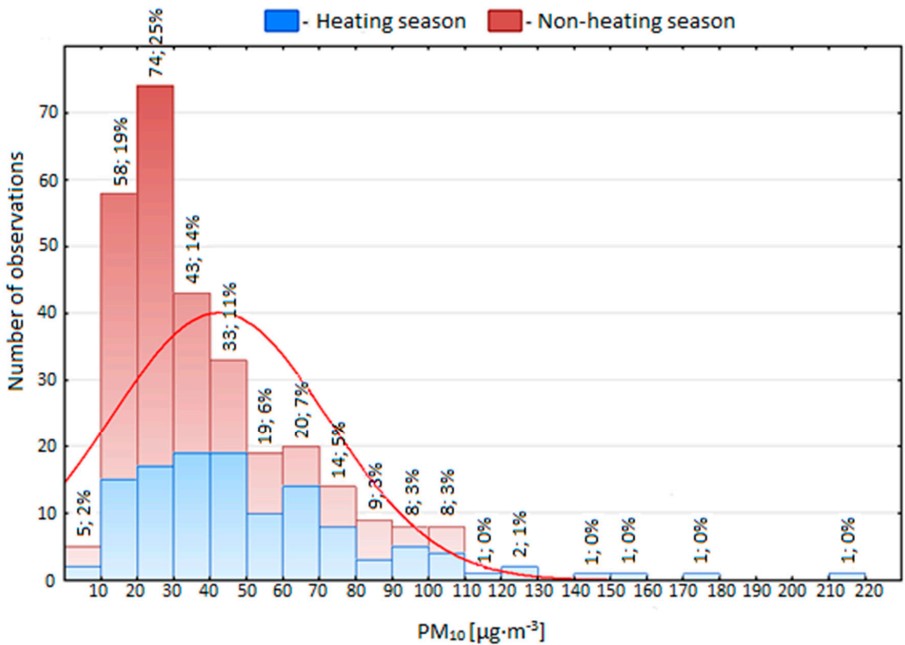

**Figure 3.** Frequency distribution of daily concentrations of particulate matter $PM_{10}$—sample point in Skawina (period: 23 February –31 December 2019).

Daily $PM_{10}$ concentrations over the entire measurement period ranged from 4.03 to 214.59 $\mu g \cdot m^{-3}$ (Table 1). The average was 42.28 $\mu g \cdot m^{-3}$, which was ~105.7% of the limit value for the annual $PM_{10}$ concentration (40 $\mu g \cdot m^{-3}$) [4]. Recorded minimum (4.03 $\mu g \cdot m^{-3}$) and maximum (214.59 $\mu g \cdot m^{-3}$) $PM_{10}$ concentration occurred during the heating season. There were significant differences in $PM_{10}$ concentrations between the research seasons (Table 1, Figures 2 and 3). However, $PM_{10}$ concentrations were relatively high in both the heating (52.68 $\mu g \cdot m^{-3}$) and non-heating (34.97 $\mu g \cdot m^{-3}$) seasons. In addition, in the entire measurement period, 85 cases (51 for heating season and 34 for non-heating one) of exceeding the limit value for daily $PM_{10}$ concentration were noted. Therefore, the condition

regarding the limit of days with exceeded limit value for daily $PM_{10}$ concentration (35 days in a calendar year) was not met.

Daily concentrations of $PM_{10}$ were described by theoretical distribution (Figure 3), with the best results obtained by using logarithmic distribution.

Based on the daily frequency histogram of $PM_{10}$ concentrations, it was found that on most days over the entire measurement period, $PM_{10}$ concentrations were in the range of 20–30 μg·m$^{-3}$ (25 and 74%, respectively). In both seasons, days with $PM_{10}$ in the range of 10–20 μg·m$^{-3}$ were also recorded relatively often (19% in the heating season and 58% in the non-heating one). Figure 3 shows a significant decrease in the number of observations with increasing $PM_{10}$ concentration. Relatively high concentrations of $PM_{10}$ (above 110 μg·m$^{-3}$) represented a small percentage in the heating season (1%), while in the non-heating season they did not occur. The occurrence of individual days with very high concentrations of $PM_{10}$ (on the order of 200 μg·m$^{-3}$) suggests the impact of episodic emissions.

High levels of $PM_{10}$ in the heating season were due to increased activity of local emission sources and a set of adverse meteorological conditions (e.g., low air temperatures, low wind speeds, limited amount of precipitation) (Figure 4). Unfavorable atmospheric conditions made it difficult to scatter and remove air pollution. To assess the impact of meteorological conditions on the daily $PM_{10}$ concentrations, a correlation matrix was prepared (Table 2).

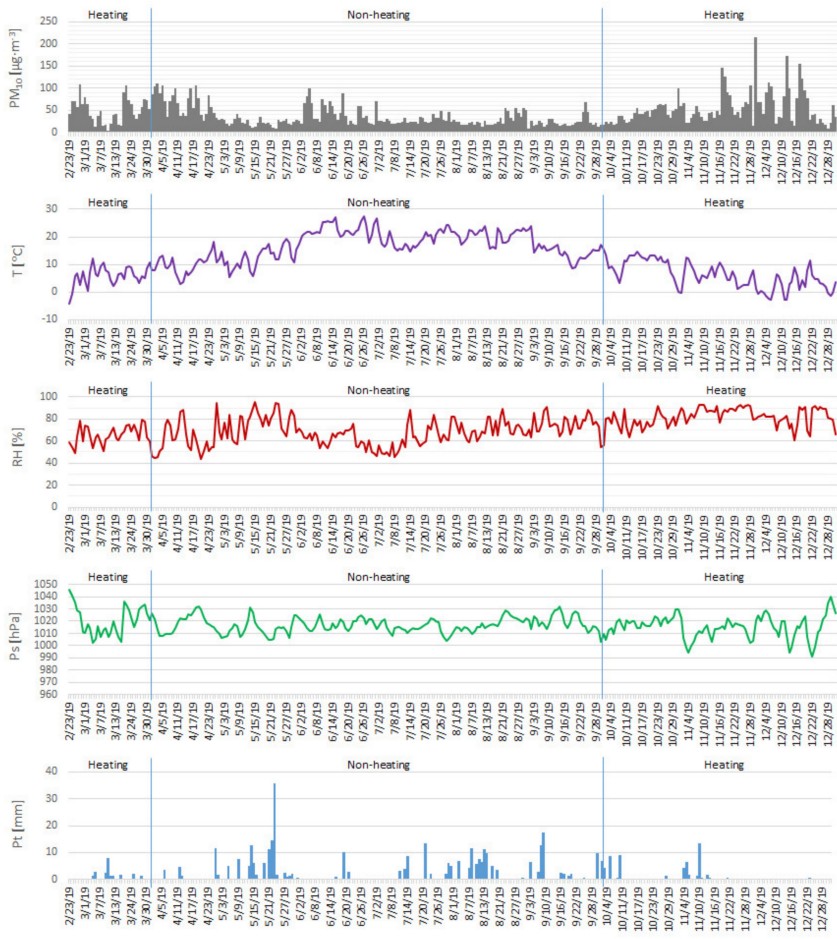

**Figure 4.** Variability of average daily $PM_{10}$ concentrations depending on meteorological parameters in the measurement period from 23 February to 31 December 2019 (Designations: T—air temperature; RH—relative humidity; Ps—atmospheric pressure; Pt—atmospheric precipitation).

**Table 2.** Nonparametric (Spearman) correlation matrix ($\alpha = 0.05$) between $PM_{10}$ concentrations ($\mu g \cdot m^{-3}$) and meteorological parameters (period: 23 February–31 December 2019).

| | T (°C) | RH (%) | Ws (m·s$^{-1}$) | Ps (hPa) | Pt (mm) |
|---|---|---|---|---|---|
| $PM_{10}$ ($\mu g \cdot m^{-3}$) | **−0.30** | 0.02 | −0.06 | **0.18** | **−0.39** |

Bold type indicates that the correlation is statistically significant ($\alpha = 0.05$).

The highest correlation was observed between $PM_{10}$ and precipitation ($r = -0.39$) and air temperature ($r = -0.30$). This means that along with the increase in precipitation and air temperature, relatively low concentrations of $PM_{10}$ occurred. Low correlation was noted for $PM_{10}$ concentrations and atmospheric pressure ($r = 0.18$). In the case of $PM_{10}$ and two other parameters (humidity and wind speed), the correlations were not statistically significant ($r = 0.02$ and $r = -0.06$, respectively).

### 3.1. Concentration of Selected Heavy Metals in $PM_{10}$

Table 3 shows the weekly concentrations of selected heavy metals in $PM_{10}$ in the entire measurement period and the heating/non-heating season. A graphic presentation of the results obtained is shown in Figure 5.

**Table 3.** Concentration of selected heavy metals ($ng \cdot m^{-3}$) in weekly samples of $PM_{10}$ from Skawina (period: 23 February—31 December 2019).

| Metals Concentration [ng·m$^{-3}$] | Entire Measuring Period | | | Heating Season | | | Non-Heating Season | | |
|---|---|---|---|---|---|---|---|---|---|
| | Ave ± SD | Min | Max | Ave ± SD | Min | Max | Av ± SD | Min | Max |
| As | 0.85 ± 0.68 | 0.33 | 4.00 | 1.29 ± 0.89 | 0.41 | 4.00 | 0.56 ± 0.24 | 0.33 | 1.34 |
| Cd | 0.62 ± 0.41 | 0.19 | 2.43 | 0.76 ± 0.47 | 0.26 | 2.43 | 0.52 ± 0.35 | 0.19 | 1.52 |
| Co | 0.68 ± 0.48 | 0.13 | 2.50 | 0.78 ± 0.38 | 0.15 | 1.40 | 0.61 ± 0.54 | 0.13 | 2.50 |
| Cr | 7.39 ± 5.25 | 2.41 | 27.62 | 6.86 ± 3.01 | 2.41 | 15.15 | 7.75 ± 6.35 | 2.61 | 27.62 |
| Cu | 11.90 ± 7.94 | 4.21 | 45.17 | 17.47 ± 9.62 | 5.19 | 45.17 | 8.18 ± 3.21 | 4.21 | 21.36 |
| Ni | 2.25 ± 1.27 | 0.88 | 7.13 | 2.51 ± 1.15 | 1.14 | 5.72 | 2.08 ± 1.34 | 0.88 | 7.13 |
| Zn | 39.49 ± 23.87 | 12.29 | 114.50 | 52.98 ± 20.94 | 18.15 | 107.97 | 30.49 ± 21.62 | 12.29 | 114.50 |
| Al | 425.57 ± 198.45 | 152.74 | 998.06 | 399.01 ± 114.59 | 170.10 | 561.14 | 443.28 ± 239.27 | 152.74 | 998.06 |
| Pb | 11.53 ± 9.17 | 2.07 | 35.77 | 18.10 ± 9.97 | 5.98 | 35.77 | 7.14 ± 5.25 | 2.07 | 21.67 |
| V | 0.95 ± 0.53 | 0.52 | 2.44 | 1.04 ± 0.44 | 0.52 | 1.76 | 0.90 ± 0.59 | 0.52 | 2.44 |
| Sum of metals | 501.23 ± 219.90 | 200.88 | 1170.63 | 500.80 ± 147.26 | 211.80 | 733.67 | 501.52 ± 260.10 | 200.88 | 1170.63 |

In the entire measurement period, the highest concentrations were observed for aluminum (Al) and zinc (Zn) (Figure 5c), and the lowest for cadmium (Cd) and carbon monoxide (CO) (Figure 5a). Generally, in the entire measuring period, the metals analyzed in $PM_{10}$ can be ordered in the following order: Al (425.57 ng·m$^{-3}$) > Zn (39.49 ng·m$^{-3}$) > Cu (11.90 ng·m$^{-3}$) > Pb ( 11.53 ng·m$^{-3}$) > Cr (7.39 ng·m$^{-3}$) > Ni (2.25 ng·m$^{-3}$) > V (0.95 ng·m$^{-3}$) > As (0.85 ng·m$^{-3}$) > Co (0.68 ng·m$^{-3}$) > Cd (0.62 ng·m$^{-3}$) (Table 2). The total share of the analyzed elements in $PM_{10}$ concentrations was ~1.37% and did not show a clear seasonal variation (~1.09% (heating season) and ~1.55% (non-heating season)). However, higher concentrations of these elements were recorded during the heating season compared to the non-heating one. The highest concentration increase during the heating season occurred in the case of Pb (61%) (Figure 5b). A significant increase was also recorded for As (57%), Cu (53%), Zn (43%) and Cd (32%). For the remaining substances, the increase during the heating season did not constitute more than 30%. For aluminum and chromium, it was 11 and 13%, respectively; however, higher concentrations of both metals occurred in the non-heating season.

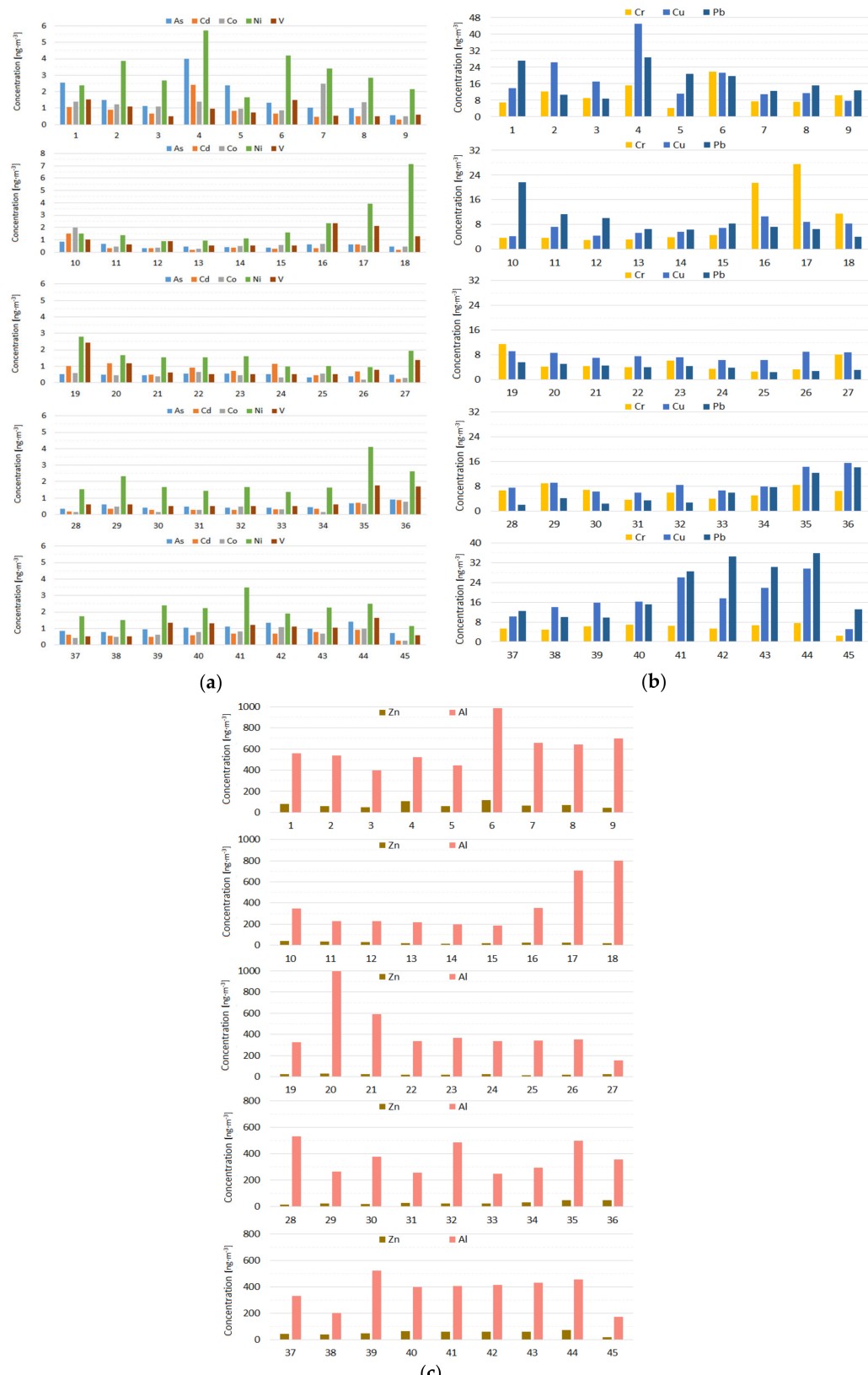

**Figure 5.** Concentration of arsenic (As), cadmium (Cd), cobalt (Co), nickel (Ni) and vanadium (V) (**a**), chromium (Cr), copper (Cu), lead (Pb) (**b**), Zinc (Zn) and aluminum (Al) (**c**) concentration (ng·m$^{-3}$) in weekly samples of PM$_{10}$ dust from Skawina (period: 23 February–31 December 2019).

### 3.2. Concentrations of Selected Polycyclic Aromatic Hydrocarbons (PAHs) in PM$_{10}$

Table 4 shows the weekly concentrations of selected polycyclic aromatic hydrocarbons (PAHs) in PM$_{10}$ in the entire measurement period and the heating/non-heating season. A graphic presentation of the results obtained is shown in Figure 6.

**Table 4.** Concentration of selected polycyclic aromatic hydrocarbons (PAHs) (ng·m$^{-3}$) in weekly samples of PM$_{10}$ from Skawina (period: 23 February–31 December 2019).

| PAHs Concentration [ng·m$^{-3}$] | Entire Measuring Period | | | Heating Season | | | Non-Heating Season | | |
|---|---|---|---|---|---|---|---|---|---|
| | Ave ± SD | Min | Max | Ave ± SD | Min | Max | Ave ± SD | Min | Max |
| Naph | 6.14 ± 12.90 | 0.12 | 46.71 | 9.38 ± 15.41 | 0.34 | 45.52 | 3.99 ± 10.69 | 0.12 | 46.71 |
| Acy | 0.68 ± 0.95 | 0.10 | 3.90 | 0.96 ± 1.11 | 0.10 | 3.90 | 0.50 ± 0.80 | 0.12 | 3.73 |
| Ace | 1.78 ± 3.51 | 0.10 | 13.27 | 2.60 ± 4.12 | 0.10 | 12.95 | 1.23 ± 2.99 | 0.12 | 13.27 |
| Fl | 2.26 ± 3.72 | 0.12 | 15.11 | 3.18 ± 4.20 | 0.63 | 14.45 | 1.64 ± 3.30 | 0.12 | 15.l1 |
| Phen | 7.94 ± 14.80 | 0.12 | 55.38 | 12.78 ± 16.64 | 0.77 | 55.38 | 4.71 ± 12.75 | 0.12 | 54.76 |
| An | 1.90 ± 3.14 | 0.12 | 11.99 | 2.91 ± 3.58 | 0.42 | 11.00 | 1.23 ± 2.67 | 0.12 | 11.99 |
| Fluo | 3.79 ± 4.66 | 0.40 | 21.36 | 7.46 ± 5.29 | 1.43 | 21.36 | 1.34 ± 1.74 | 0.40 | 6.62 |
| Pyr | 4.50 ± 6.29 | 0.28 | 27.48 | 9.53 ± 7.36 | 1.52 | 27.48 | 1.16 ± 1.53 | 0.28 | 5.70 |
| BaA | 6.75 ± 12.03 | 0.10 | 49.88 | 15.52 ± 15.33 | 1.99 | 49.88 | 0.91 ± 1.64 | 0.10 | 6.15 |
| Chry | 4.27 ± 5.29 | 0.51 | 21.15 | 8.85 ± 5.80 | 2.61 | 21.15 | 1.21 ± 1.10 | 0.51 | 4.72 |
| BbF | 5.84 ± 6.93 | 0.91 | 26.65 | 11.79 ± 7.76 | 2.43 | 26.65 | 1.88 ± 1.10 | 0.91 | 4.81 |
| BkF | 2.63 ± 2.15 | 0.69 | 8.25 | 4.51 ± 2.25 | 1.61 | 8.25 | 1.38 ± 0.65 | 0.69 | 3.37 |
| BaP | 4.26 ± 4.36 | 1.00 | 16.80 | 8.11 ± 4.63 | 2.61 | 16.80 | 1.70 ± 1.06 | 1.00 | 5.07 |
| IcdP | 3.31 ± 3.43 | 0.70 | 13.79 | 6.30 ± 3.70 | 1.83 | 13.79 | 1.32 ± 0.79 | 0.70 | 3.67 |
| DahA | 1.11 ± 0.70 | 0.12 | 3.26 | 1.59 ± 0.84 | 0.27 | 3.26 | 0.78 ± 0.32 | 0.12 | sty.37 |
| BghiP | 2.97 ± 2.88 | 0.83 | 11.61 | 5.49 ± 3.14 | 1.44 | 11.61 | 1.28 ± 0.54 | 0.83 | 2.90 |
| Sum of PAHs | 60.14 ± 67.17 | 9.00 | 237.91 | 110.96 ± 66.82 | 25.84 | 237.91 | 26.26 ± 41.65 | 9.00 | 177.33 |

Designations: Naph—naphthalene; Acy—acenaphthylene; Ace—acenaphten; Fl—fluorene; Phen—phenanthrene; An—anthracene; Fluo—fluoranthene; Pyr—pyrene; BaA—benzo[a]anthracene; Chry—chrysen; BbF—benzo[b]fluoranthene; BkF—benzo[k]fluoranthene; BaP—benzo[a]pyrene; IcdP—indeno[123-cd]pyrene; DahA—dibenzo[ah]anthracene; BghiP—benzo[ghi]perylene.

Chemical composition analyzed have shown the presence of polycyclic aromatic hydrocarbons (Table 3, Figure 6). The share of the analyzed PAHs in PM$_{10}$ concentrations was ~1.25% and was higher in the heating season (~2.19%) compared to the non-heating one (~0.63%). Analyses have shown particularly high concentrations for several PAHs. The maximum value over the entire measuring period was recorded for Phen (55.38 ng·m$^{-3}$) (Figure 6a). High concentrations were also reported in the case of BaA and Naph (49.88 and 46.71 ng·m$^{-3}$, respectively) (Figure 6b,d). Relatively high concentrations of benzo[a]pyrene were recorded, which ranged from 1.00 to 16.80 ng·m$^{-3}$ (Figure 6c). The average concentration of BaP in the entire measuring period was 4.26 ng·m$^{-3}$. This indicates a potentially high health risk for the inhabitants of the research area resulting from exposure to PAHs, especially in the winter season. Among all PAHs, the largest difference of concentrations between the heating (8.11 ng·m$^{-3}$) and non-heating (1.70 ng·m$^{-3}$) season occurred in the case of BaP. Similar seasonal differences were also observed for the remaining PAHs analyzed. A significant increase in concentrations during the heating season was observed for Fluo, Pyr, Chry, BbF, BaP, IcdP, BghiP (~80%). For other substances, the increase was also high (above 48%).

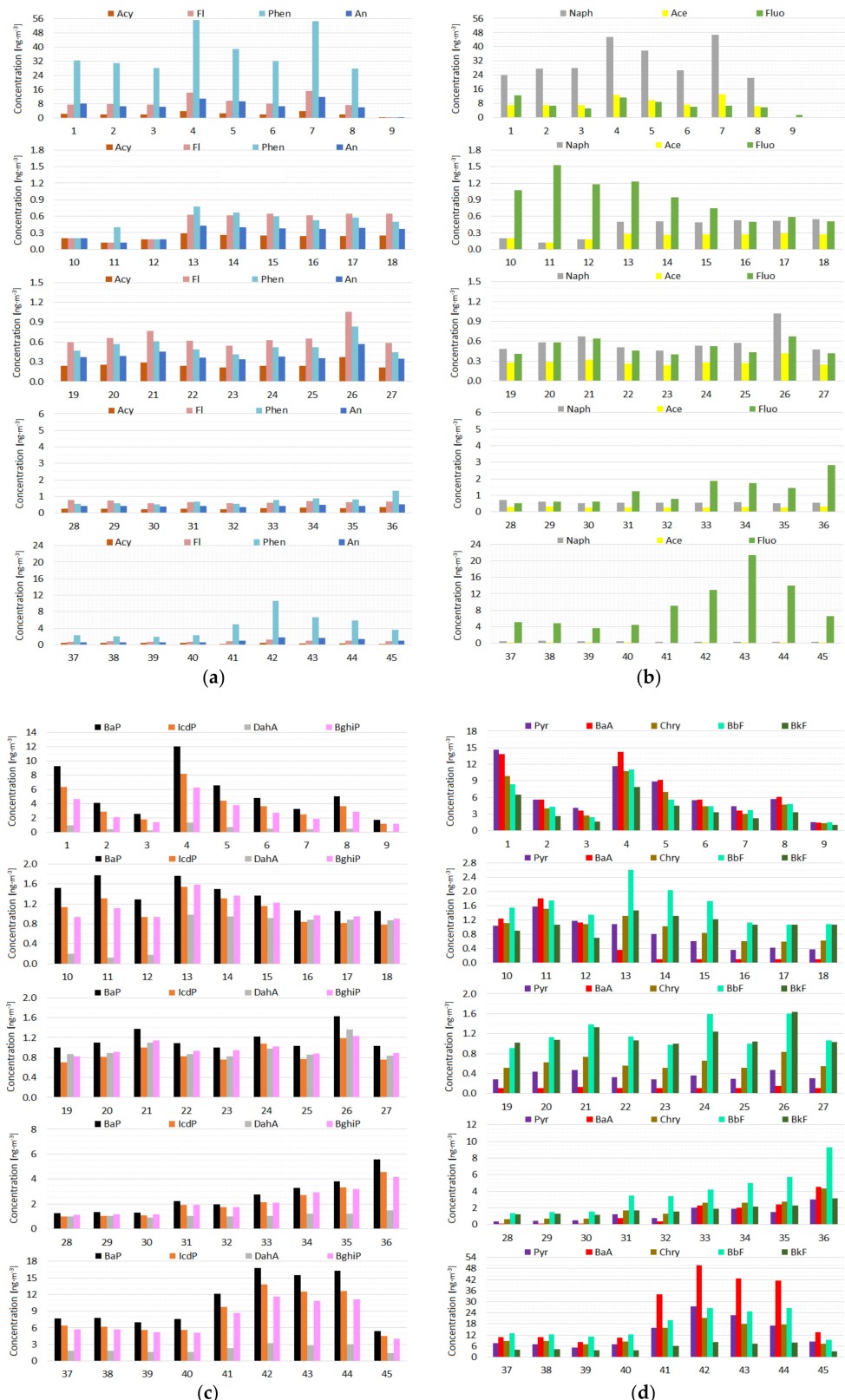

**Figure 6.** Concentration of selected PAHs (ng·m$^{-3}$): acenaphthylene, fluorene, phenanthrene, anthracene (**a**), naphthalene, acenaphten, fluoranthene (**b**), benzo[a]pyrene, indeno[123-cd]pyrene, dibenzo[a,h]anthracene, benzo[ghi]perylene (**c**), pyrene, benzo[a]anthracene, chrysene, benzo[b]fluoranthene, benzo[k]fluoranthene (**d**) in weekly samples of PM$_{10}$ dust from Skawina (period: 23 February–31 December 2019).

## 4. Discussion

The results of measurements of daily concentrations of $PM_{10}$ and its contained metals and polycyclic aromatic hydrocarbons (PAHs) are presented in Chapter 3. The $PM_{10}$ fraction is a parameter commonly used to assess air quality in Europe [1]. In this study, $PM_{10}$ concentrations with metals and PAHs were also used to assess air quality in southern Poland.

Based on the obtained data, it was found that $PM_{10}$ concentrations showed a clear seasonal variation. In the heating season, increased concentrations of $PM_{10}$ (52.68 $\mu g \cdot m^{-3}$) were observed compared to the non-heating season (34.97 $\mu g \cdot m^{-3}$). Seasonal differences are a common phenomenon occurring at urban and industrial stations in Poland [24,25] and other countries [26].

The average concentration of $PM_{10}$ in the entire measurement period was 42.28 $\mu g \cdot m^{-3}$. Similar concentrations of $PM_{10}$ (40 $\mu g \cdot m^{-3}$) were observed in 2007 and 2010 in Varna [27].

Both in the heating and non-heating season there were cases of exceeding the limit value for daily $PM_{10}$ (51 and 34 cases, respectively). The analysis of the air quality assessments carried out for the years 2003–2013 indicates that the air quality condition in Poland is systematically improving. In urban agglomerations in Poland, the average annual $PM_{10}$ concentration in Poland decreased, but the EU limit value is still being exceeded (40 $\mu g \cdot m^{-3}$) [28]. Based on $PM_{10}$ concentrations obtained for 229 air monitoring points, it was found that for 127 of them the concentrations were inconsistent with the average EU daily limit value (50 $\mu g \cdot m^{-3}$). In addition, research conducted in 2016 showed that the highest concentrations of $PM_{10}$ in Poland occurred in the agglomeration of Katowice, Rybnicko-jastrzębska and Kraków [29]. Elevated $PM_{10}$ concentrations in Krakow have also been identified in earlier years [30].

The present research indicates that episodic emissions in Skawina (Kraków) are responsible for the formation of a relatively high level of $PM_{10}$ during the year. An example of episodic emissions is the maximum concentration of $PM_{10}$ observed on November 30, 2019 (214.59 $\mu g \cdot m^{-3}$). In addition, this is confirmed by the histogram (Figure 3), which shows a small percentage of observations of concentrations above 109.84 $\mu g \cdot m^{-3}$ (1%).

Based on Figure 3, it was found that on most days both during the heating and non-heating season, $PM_{10}$ concentrations ranged from 10–30 $\mu g \cdot m^{-3}$. Much higher concentrations of $PM_{10}$ (30-105 $\mu g \cdot m^{-3}$) dominated throughout the entire measurement period (from September 2015 to August 2016) in three cities in China: Chongqing, Wuhan, and Nanjing [31]. In addition, the results of the obtained $PM_{10}$ concentrations in Skawina agree well with the results of the logarithmic fit. High quality matching of the logarithmic distribution to empirical data ($PM_{10}$) has also been identified in studies conducted in Manchester and Athens [32].

Relatively high concentrations of $PM_{10}$ recorded in the non-heating season (average: 34.97 $\mu g \cdot m^{-3}$) indicated that mobile and industrial sources as well as the processes of secondary dusting of road and/or soil dust could have played an important role. High concentrations of $PM_{10}$ during the heating season (average: 52.68 $\mu g \cdot m^{-3}$) may be caused by increased activity of local emission sources and a set of adverse meteorological conditions. To assess the impact of meteorological conditions, the correlations between $PM_{10}$ and air temperature, relative humidity, atmospheric pressure and atmospheric precipitation were checked.

There was a strong correlation between $PM_{10}$ and precipitation ($r = -0.39$) and air temperature ($r = -0.30$). Precipitation directly affects $PM_{10}$ concentrations, causing it to be removed from the atmosphere [33]. Low air temperatures were associated with the intensification of combustion of solid fuels, biofuels and biomass for heating purposes, which indirectly affected the increase in $PM_{10}$ concentrations [34]. A similar correlation was noted between $PM_{10}$ and air temperature ($r = -0.26$) at the industrial air monitoring station in Târgovişte [35]. A strong correlation of $PM_{10}$ concentrations and air temperature was also found during research in the city of Elche in Spain [36]. The results show that high episodes of $PM_{10}$ concentrations in the winter season in Poland were caused both by emissions from anthropogenic sources and adverse meteorological conditions [7].

High concentrations of $PM_{10}$ recorded each year indicate that chemical composition analyzes are necessary to identify dominant sources of pollution.

The results obtained as part of the work will enhance the existing knowledge about the status of heavy metals and PAHs in urban areas. Understanding the origin of heavy metals and PAHs can be the basis for urban planning and appropriate reduction of emissions from sources.

In the entire measurement period, the largest share in $PM_{10}$ was recorded for Al (425.57 ng·m$^{-3}$) and Zn (39.49 ng·m$^{-3}$). Al and Zn have been identified as the dominant elements in $PM_{10}$ also at the urban-industrial station in Acerra [37]. It was noted that both in Skawina and Acerra, Al occurred in higher concentrations during the non-heating season. In Skawina, in the heating season, the Zn concentration was 52.98 ng·m$^{-3}$. Much lower Zn concentration (41.5 ng·m$^{-3}$) in the winter season was recorded at another station in the city of Torino in Italy [38]. In this work, higher concentrations in the non-heating season were also noted for Cr, while the other elements were at higher levels in the heating season. During the heating season, the Cr concentration was 6.86 ng·m$^{-3}$ and was much higher compared to the station in Torino [0.4 ng·m$^{-3}$] [38]. Cr concentration increased during the non-heating season (7.75 ng·m$^{-3}$), however it was lower than at the station located in Moravian-Silesian Region (13. 57 ng·m$^{-3}$) [39]. The Cr concentration in the entire measuring period (7.39 ng·m$^{-3}$) was higher compared to the values recorded at other measuring stations in Poland (e.g., Zabrze (1.7 ng·m$^{-3}$), Łódź (3.3 ng·m$^{-3}$), Warsaw (1.2 ng·m$^{-3}$)) [40] and lower than at the station in Frankfurt, Germany (17.0 ng·m$^{-3}$) [41].

High concentrations of Al and Cr throughout the measurement period indicate the presence of strong anthropogenic sources of these elements. In the case of Al, the most likely source of emissions is the processing of aluminum scrap in Skawina, while the main source of Cr emissions is the metallurgy, refractory, chemical and tanning industries [42].

The concentrations of metals covered by legal regulations (As (0.85 ng·m$^{-3}$), Cd (0.62 ng·m$^{-3}$), Ni (2.25 ng·m$^{-3}$) and Pb (11.53 ng·m$^{-3}$)) in entire measurement period were relatively low and did not exceed the permissible (Pb: 500 ng·m$^{-3}$) and target values (As, Cd and Ni: 6, 5 and 20 ng·m$^{-3}$, respectively) [4]. A similar level of Pb (11.0 ng·m$^{-3}$) was recorded in Frankfurt, Germany [41], while As and Cu concentration at the same measuring station was higher (1.7 and 105 ng·m$^{-3}$, respectively). Metals such as Zn, Cr, Cu, As and Pb were measured in Krakow (Nowa Huta) in earlier years [43]. The results from the heating seasons of 2009 and 2010 showed significantly higher concentrations of Zn (442 ng·m$^{-3}$) compared to the results from Skawina (52.98 ng·m$^{-3}$). Definitely higher concentrations in the heating season 2009 and 2010 were also recorded in the case of Cr, Cu, As and Pb.

Chemical composition analyzes have shown the presence of polycyclic aromatic hydrocarbons from incomplete combustion of fossil fuels and other substances in municipal and industrial processes [44]. Relatively high concentrations of benzo[a]pyrene, considered a carcinogen, have been reported. The average concentration of BaP over the entire measurement period was 4.26 ng·m$^{-3}$, four times higher than the target value for the average annual concentration of BaP (1 ng·m$^{-3}$) [4]. The most probable source of BaP emissions was the combustion of fuels in households, which is indicated by the clear difference between the BaP concentrations recorded in the heating season (average: 8.11 ng·m$^{-3}$) and these from the non-heating season (average: 1.70 ng·m$^{-3}$). Similar seasonal differences were also observed for the remaining PAHs analyzed. The DahA concentration, with a carcinogenicity factor five times higher compared to BaP, remained at a relatively low level (1.11 ng·m$^{-3}$, the whole period).

In the entire measuring period, the maximum concentration was recorded in the case of Pyr (7.94 ng·m$^{-3}$), and it was definitely higher than at urban-industrial station in Athens in Greece (0.2 ng·m$^{-3}$) [45]. At the same station, lower concentrations were also recorded for An (0.21 ng·m$^{-3}$), Fl (0.22 ng·m$^{-3}$) and Chry (0.44 ng·m$^{-3}$) compared to Skawina (1.90, 2.26 and 4.27 ng·m$^{-3}$, respectively). Lower concentrations of Pyr (0.15 ng·m$^{-3}$), An (0.01 ng·m$^{-3}$) and Chry (0.09 ng·m$^{-3}$) were recorded in studies conducted in Spain [46]. In this case, the concentrations of substances such as: BkF (0.07 ng·m$^{-3}$), BaP (0.04 ng·m$^{-3}$), BaA (0.04 ng·m$^{-3}$), BbF (0.10 ng·m$^{-3}$) were also lower than in Skawina (2.63, 4.26, 6.75, 5.84 ng·m$^{-3}$, respectively)

The impact of coal and biomass combustion [47] on PAHs air pollution confirms the ratio of fluoranthene concentration to the sum of fluoranthene and pyrene concentrations (0.51). The ratio of benzo[a]anthracene concentration to the sum of benzo[a]anthracene and chrysene concentration (average: 0.39) also suggests a large role of combustion processes. In the non-heating season the ratio was lower (average: 0.26), which indicates the communication sector as the source of emissions. The share of communication sources also confirms the presence of dibenzo[a,h]anthracene, often found in the vicinity of communication routes [48].

## 5. Conclusions

Measurements of $PM_{10}$ and associated heavy metals and polycyclic aliphatic hydrocarbons were carried out from 23 February to 31 December 2019 in Skawina (southern Poland). The obtained results indicated very high $PM_{10}$ concentrations in the analyzed area. In the whole measurement period, 85 cases of exceeding the limit value for daily $PM_{10}$ concentration were noted (51 cases in the heating season and 34 in the non-heating one).

The presence of polycyclic aromatic hydrocarbons was associated with the processes of incomplete combustion of fuels and other flammable substances of unknown origin. The variability of the concentrations of individual compounds from the PAHs group pointed to the dominant role of emissions from the combustion of fossil fuels and biomass in municipal sources. This factor could significantly contribute to the deterioration of air quality in the study area, which is indicated by the high concentration of benzo[a]pyrene in relation to the target value, especially during the heating season. High and comparable concentrations of aluminum in both seasons indicate the presence of substances from aluminum treatment (Al, Zn) in $PM_{10}$ dust. The concentrations of metals covered by legal regulations (As, Cd, Ni and Pb) were relatively low and did not exceed the permissible and target values.

**Author Contributions:** Conceptualization, formal analysis, visualization, formal analysis, writing original draft preparation, N.Z.; methodology, validation, writing—review and editing, data curation, K.S., project administration, formal analysis, supervision N.Z and K.S. All authors have read and agree to the published version of the manuscript.

**Funding:** The research was co-financed by town and municipal office in Skawina.

**Acknowledgments:** The authors would like to thank the town and municipal office in Skawina for co-financing the project.

**Conflicts of Interest:** The authors declare no conflict of interest.

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
