# Peer review of "The Content of Selected Heavy Metals and Polycyclic Aromatic Hydrocarbons (PAHs) in PM10 in Urban-Industrial Area"

_sustainability, doi:10.3390/su12135284_

Round 1

Reviewer 1 Report

Interesting article concerning the measurement of PM10 concentrations and the assessment of heavy metals and polycyclic aromatic hydrocarbons (PAHs).

Some recommendations are indicated in order to increase the value of the paper

  • more information concerning the other sources of heavy metals, PAH... as the smoking status of populations, alimentation, working in different media... could be introduced, also because of the interferences with pollution data of the area - correlations with health status are welcome
  • tables are necessary to be formatted as the journal recommendations - see table 3, table 4,
  • the quality of Fig 6 needs to be improved or to be considered as supplementary data...
  • the discussion part needs to be improved with more comparison with other data, underlining the originality and new added data of this article

Reviewer 2 Report

Dear Authors,

The paper entitled "The Content of selected heavy metals and polycyclic 2 aromatic hydrocarbons (PAHs) in PM10 in 3 urban-industrial area" is based on a campaign of measures of PM10 in a region of Poland. The topic responds to criteria of the Journal Sustainability. The paper needs and Extensive editing of English language. Several errors or missprints are presented in all parts of the paper and make it barely readable. I would suggest to contact a mother-tongue eneglish reviewer and resubmit it. 

Reviewer 3 Report

Comments :

Recommendation Regarding This Manuscript:

The manuscript reports a case study of daily PM10 concentrations and the assessment of heavy metals and polycyclic aromatic hydrocarbons (PAHs) carried out particularly in the industrial-urban area in southern Poland in the period from February to December 2019, covering two seasons: heating and non-heating. They have used Mass Spectrometry with Inductively Excited Plasma ICP-MS for the determination of metal content – As, Cd, Pb, Ni, Co, Cr, Cu, Zn, V, Atomic Emission Spectrometry with Induced Plasma ICP-OES for Au and Mg. Analysis of selected PAHs (Naph, Acy, Ace, Fl, Phen, An, Fluo, Pyr, BaA, Chry, BbF, BkF, BaP, IcdP, DahA, BghiP) was performed using a Gas Chromatography coupled with Mass Spectrometry GC-MS. The authors have reported a standard experimental design, results, discussion and conclusion. The overall research/study and the way of presentation do justice to the manuscript. However, the manuscript can be accepted for publication, only after going through following minor revisions.

Comments:

1)The English and grammar should be checked throughout the whole manuscript.

2) The author should describe what heating and non-heating actually means, and why it is necessary to cover these two seasons.

3) In the experimental section, is it possible to elaborate the samples type and method of collection more elaborately, The authors have mentioned only (the collection was made….what is the nature and state of collection?).

4) The figures especially 5 and 6 are too small to read. It is understandable that the authors have worked very hard to collect the data, but a clear presentation is also necessary for the readers.

5) In the conclusion section, if the authors present a comparison of the result with the global scenario that would be better that just limiting the data on Skawina. It seems more like a report to be submitted to a city office.

The manuscript can be accepted after revising all above minor points.

Round 2

Reviewer 1 Report

The authors answered to my requests, so I consider that the article is suitable for publication

Reviewer 2 Report

The paper can be accepted after the first revision